# Real-time Core-Periphery Guided ViT with Smart Data Layout Selection on Mobile Devices

**Zhihao Shu**[1][*] **Xiaowei Yu**[2][*] **Zihao Wu**[1] **Wenqi Jia**[2] **Yinchen Shi**[3]
**Miao Yin**[2] **Tianming Liu**[1] **Dajiang Zhu**[2] **Wei Niu**[1]
[1]University of Georgia    [2]University of Texas at Arlington    [3]NYU
{Zhihao.Shu, Zihao.Wu1, wniu}@uga.edu
tliu@cs.uga.edu
ys4653@nyu.edu
{xxy1302, wxj1489}@mavs.uta.edu
{dajiang.zhu, miao.yin}@uta.edu

## Abstract

Mobile devices have become essential enablers for AI applications, particularly in scenarios that require real-time performance. Vision Transformer (ViT) has become a fundamental cornerstone in this regard due to its high accuracy. Recent efforts have been dedicated to developing various transformer architectures that offer improved accuracy while reducing the computational requirements. However, existing research primarily focuses on reducing the theoretical computational complexity through methods such as local attention and model pruning, rather than considering realistic performance on mobile hardware. Although these optimizations reduce computational demands, they either introduce additional overheads related to data transformation (e.g., Reshape and Transpose) or irregular computation/data-access patterns. These result in significant overhead on mobile devices due to their limited bandwidth, which even makes the latency worse than vanilla ViT on mobile. In this paper, we present ECP-ViT, a real-time framework that employs the core-periphery principle inspired by the brain functional networks to guide self-attention in ViTs and enable the deployment of ViT models on smartphones. We identify the main bottleneck in transformer structures caused by data transformation and propose a hardware-friendly core-periphery guided self-attention to decrease computation demands. Additionally, we design the system optimizations for intensive data transformation in pruned models. ECP-ViT, with the proposed algorithm-system co-optimizations, achieves a speedup of $4.6\times$ to $26.9\times$ on mobile GPUs across four datasets: STL-10, CIFAR100, TinyImageNet, and ImageNet.

## 1 Introduction

In recent decades, there has been a significant increase in applying deep neural network (DNN) architectures across various fields, including autonomous driving [23], natural language processing [9], extended reality (XR) [13], image processing [22], and View Synthesis [32]. Along with significant progress in hardware performance and public datasets, more and more complex DNN architectures have been proposed, including Convolution Neural Network [52], RNN [36], Transformer [51]. These architectures have led to groundbreaking breakthroughs in various application domains by leveraging their powerful feature extraction abilities. Meanwhile, mobile devices have become essential for deploying these applications, especially in scenarios that demand real-time performance. The mobile GPU on Oneplus 11 can achieve a theoretical *peak performance of over 3T FLOPS* (floating point

---

[*]Equal contributions.

38th Conference on Neural Information Processing Systems (NeurIPS 2024).

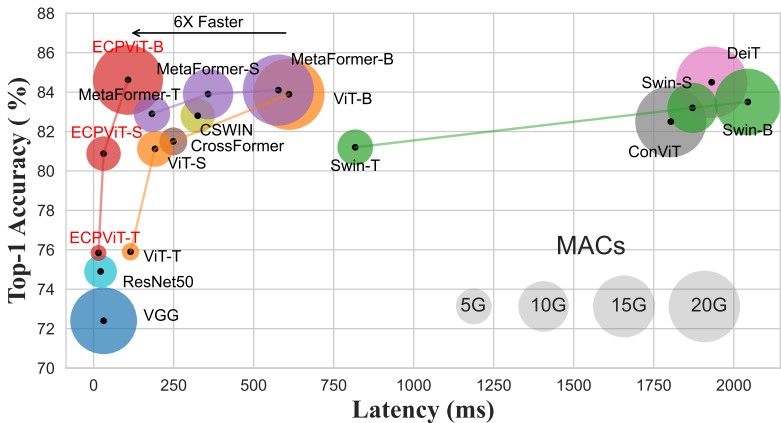

Figure 1: Accuracy-Latency-MACs comparison among various models on ImageNet-1K. Latency is measured on Oneplus 11 (Snapdragon 8 Gen 2 SoC). The radius of each circle represents the model's MACs (Multiply-Accumulate Operations). Our ECP-ViT-Base achieves the highest top-1 accuracy and offers the fastest inference speed among transformer-based models, providing 6× faster than ViT-Base. We simplify the notation as "Base" = B, "Small" = S, and "Tiny" = T.

operations per second) [1]. Their widespread availability and increasing computational capabilities make them ideal platforms for extending the impact of these algorithm innovations.

Compared to traditional convolutional neural networks (CNNs), transformers have recently become widely used across various areas and tasks due to their high accuracy. The attention mechanism in the transformer allows neurons to exchange messages effectively and efficiently, leading to promising results in natural language processing [48, 16] and computer vision domains [17, 53]. However, transformer architectures also known for their deeper network layers and require frequent reshaping and transposing of the feature maps. This results in more intermediate results and leads to a memory bound for the computation. Optimizing transformers for efficient execution becomes particularly challenging in environments where memory bandwidth is limited.

Advancements in transformer architecture design, including Vision Transformer (ViT) [17], have focused on improving message exchange mechanisms among spatial tokens through different Token Mixers. Other efforts include the shifted window attention in Swin [34], the token-mixing MLP in Mixer [44], and the pooling in MetaFormer [57]. These designs aim to enhance self-attention accuracy and hardware efficiency compared to the original vanilla ViT [17], enabling more effective and efficient execution on various hardware platforms. Despite the significant progress made in transformer architecture, particularly in reducing theoretical computation demands, achieving real-time performance for transformer-based models on mobile devices remains a major challenge. Figure 1 illustrates the latency, accuracy, and floating-point operations (FLOPs) comparison of different models on a mobile GPU. Despite having fewer FLOPs, Swin-T (local attention) is even more than 3× slower than ViT-T (global attention). This is because local attention requires more frequent data transformation to reorganize tokens, which accounts for 69% of execution time in Swin. In contrast, these numbers are 43% in ViT and only 0.8% in VGG16. Similarly, DeiT has a broadly similar structure to ViT, but it involves more complex data layout transformations and larger intermediate results, as illustrated in Table 1. Unlike powerful server platforms, mobile devices have limited memory bandwidth [24], making it challenging to benefit from reduced theoretical FLOPs. There has been a fundamental lack of general principles that involve co-designing model architecture and system optimizations.

In this work, we present an integrated framework that incorporates co-optimizations by revisiting the model design and system optimizations. First, we propose a hardware-efficient and computational-friendly sparse scheme (guided by the Core-Periphery principle in brain networks) [61, 59, 60, 58, 62] that can be applied to guide the message exchange in self-attention. This scheme also helps reduce bandwidth pressure for the subsequent Softmax operation. Secondly, we develop a set of comprehensive compiler-based optimizations **supporting the proposed pruning scheme and**

Table 1: Comparison between ViT and DeiT. All data is measured on Oneplus 11 (Snapdragon 8 Gen 2 SoC). Layout Transformation indicates the time spent on transforming the tensor's layout, such as Transpose and Reshape. Computation indicates the time spent on pure tensor computation.

| Model | Layout Transformation (ms) | Computation (ms) | Intermediate Results (MB) |
|---|---|---|---|
| ViT-Base [17] | 324 | 286 | 106.47 |
| DeiT-Base [46] | 1303 | 633 | 125.42 |

**fully eliminate the overhead for Transpose and Reshape**. Enabled by the advanced compiler optimizations, it is possible to achieve both high accuracy and high acceleration simultaneously. We demonstrate that, 1) our proposed fine-grained structured core-periphery guided self-attention offers advantages in both accuracy and speed, and 2) our compiler framework exhibits superior end-to-end acceleration performance for both the original and proposed ECP-ViT models.

We summarize our contributions in three aspects:

- We incorporate the organizational principle of brain functional networks—specifically, the core-periphery principle—to guide self-attentions in ViTs. Additionally, we introduce a compiler code generation framework that efficiently supports the core-periphery structures on mobile devices.
- We develop comprehensive compiler-based optimizations that can fully eliminate the overhead of data transformation, bridging the gap between accuracy and latency.
- We build ECP-ViT, an end-to-end framework that combines algorithm and system design to achieve real-time performance on mobile devices. ECP-ViT achieves the significant speedup on off-the-shelf mobile devices, reaching up to $16.1\times$ on ImageNet inference while maintaining an accuracy of 84.51%.

## 2 Background and Motivation

**Mixed blessing of attention.** ViTs are widely utilized as robust backbones across various tasks. The two primary types of attention are Global Attention [3, 33] and Local Attention [39, 41, 6, 42, 7]. Global Attention, as seen in standard transformer models, allows each token to attend to every other token in the input sequence. This mechanism ensures comprehensive context capture, leading to high accuracy. However, the downside is its computational intensity, as it scales quadratically with the input length, making it less efficient for longer sequences or resource-constrained platforms.

Local Attention [64, 49] restricts the focus of each token to a subset of adjacent tokens (or with specific patterns), resulting in the reduction of computational complexity. However, local Attention does not always translate into proportionate realistic speedups when compared to the theoretical reduction in floating-point operations (FLOPs). This approach significantly reduces the computational complexity but at the cost of more layout transformations. These factors offset the expected gains from reduced computational complexity, particularly in environments like mobile GPUs where memory bandwidth and efficient data handling are critical, as reflected in Figure 1. The actual performance gains need to be evaluated in the context of specific hardware and application requirements. Compared to the above two types of attention, ECP-ViT is the first work to explore a speed-aware end-to-end framework that achieves real-time performance on real-world mobile devices for ViTs.

**Efficient network design and architecture search.** In this category, a significant methodology is Neural Architecture Search (NAS) [67, 40, 18, 31]. Some of the work leverages network pruning and sparse training to further reduce the theoretical FLOPs. At the token level, Tang et al. [43] introduces a patch slimming method to remove redundant tokens. Evo-ViT [54] updates selected informative and uninformative tokens through distinct computation paths, while VTP [66] reduced embedding dimensionality by introducing control coefficients. At the model architecture level, UP-ViTs [56] adopts a unified approach to prune channels in ViTs. SViTE [10] dynamically extracts and trains sparse sub-networks instead of training the entire model. Despite the significant progress made by these methods, both token-sampling and data-driven strategies may heavily depend on specific datasets and tasks, limiting the generalization capability of vision transformers. Additionally, *these token-level pruning or selection introduce additional operations for the Reshape and Transpose to the feature map*, leading to fewer benefits from reduced computation complexity. In contrast, ECP-ViT achieves a significant speedup on mobile platforms through two key components: 1) the more efficient

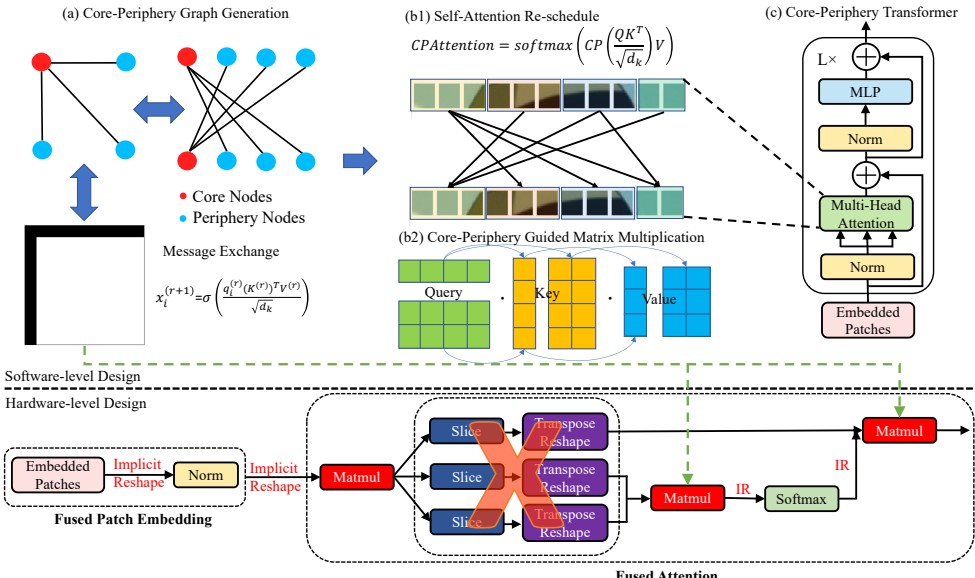

Figure 2: Co-Design of Core-Periphery Principle Guided self-attention mechanism in ECP-ViT. At the software-level design, the rescheduled interactions between patches in ECP-ViT are guided by the generated Core-Periphery (CP) graphs. Moreover, the multiplication of query, key, and value matrices in the self-attention mechanism of ECP-ViT is also under the guidance of the core-periphery graph. At the hardware-level design, the slice and transpose reshape operations are eliminated.

self-attention guided by the Core-Periphery principle, and 2) our compiler-based optimizations that eliminate data transformation overhead (i.e., `Reshape` and `Transpose`).

**DNN frameworks on mobile devices.** Recently, there has been a dedicated focus on developing inference acceleration frameworks for mobile devices from both academia and industry. Some efforts include MCDNN [21], DeepSense [55], MobiSR [30], and PatDNN [38]. However, none of these frameworks support the execution of transformer models on mobile devices. Other efforts have been made to optimize the execution of transformer models on mobile devices, include TFLite [2], TVM [11], MNN [26], Pytorch-Mobile [25], and DNNFusion [37]. They support optimizations including operator fusion, constant folding, and quantization on mobile devices. *However, they are not able to eliminate the intensive* `Reshape` *and* `Transpose` *operations in transformer models.* In this work, our goal is to find the most appropriate CP pruning scheme for mobile ViT acceleration and the corresponding full-stack acceleration framework.

## 3 Methodology

The entire framework of co-design for ECP-ViT is illustrated in Figure 2. It comprises software-level design (algorithm) and hardware-level design, both of which we will discuss in detail in the subsequent sections.

### 3.1 Problem Definition and Research Issues

As mentioned earlier, the attention mechanism in transformer models involves a large amount of data transformation, which poses significant challenges to hardware efficiency and deployment on mobile devices. This is because (i) *data transformation is memory-bound, requiring high memory bandwidth.* On mobile devices, which typically have restricted memory bandwidth, this leads to increased time consumption for processing these transformations; and (ii) data transformation in transformers, especially within the multi-head attention mechanism, *often results in irregular data access patterns.* This is typically observed during operations like tensor reorganization. Such irregular access patterns

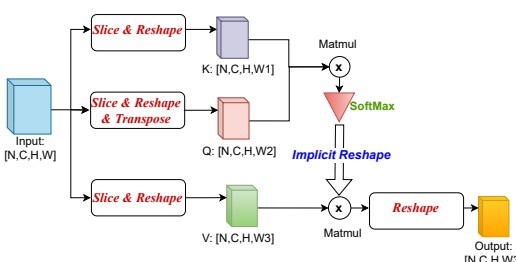

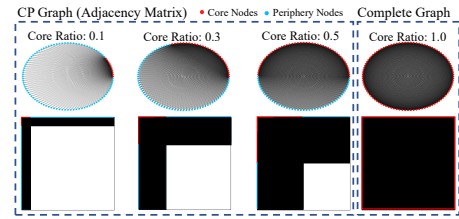

Figure 3: Illustration of ViT on attention module. In this example, we use the NCHW format for presenting our data. Words with red color are explicit data layout and transformative operators.

Figure 4: Examples of Core-Periphery Nodes in the graph, where the core ratio is calculated by dividing the number of core nodes by the total number of nodes. The white area represents pruned nodes. In adjacency matrices, black color indicates connections between nodes, while white represents no edge.

can significantly degrade performance, particularly in the absence of sufficient concurrent processing threads.

We classify the data transformation into two categories (as shown in Figure 3), `Explicit` and `Implicit`. `Explicit` transformations (red color operators in round boxes) are designed by the algorithm developer to ensure the model follows its intended computational logic. On the other hand, `Implicit` transformations adopt data layouts that best suit layout-sensitive operators for increased effectiveness. `Implicit` focuses more on performance optimization tailored to specific hardware platforms and depends on the inference framework. For instance, TFLite may prefer using NHWC layout for `MatMul` layer, while others may prefer NCHW ones. Based on the classification mentioned above, ECP-ViT specifically answers these three questions to achieve computation efficiency and data transformation elimination:

- How to design a *hardware-friendly pruning scheme* for ViTs without compromising accuracy?
- How to *effectively minimize the data transformation overhead* without affecting accuracy?
- How to flexibly *reduce the memory pressure* on mobile?

## 3.2 Core-Periphery Guided Self-Attention in ViT

The design of `ECP-ViT`, as shown in Figure 4, is inspired by Brain Neural Networks. Specifically, Figure 4 illustrates the selection of various CP graphs referenced in Figure 2 (a). The workflow involves CP graph generation, CP-guided self-attention, and CP-guided QKV multiplication, corresponding to Figure 2 (a), Figure 2 (b1), and Figure 2 (b2). In these networks, the core nodes maintain connections to all other core nodes, while edge nodes only connect to a subset (or empty) of the core nodes. We employ Grad-CAM to identify important regions of the images and assign the core nodes to those regions during training. Accordingly, the QKV matrices of these patches are divided into core and peripheral components. For example, for images with a resolution of 224x224 and a patch size of 16x16, there are a total of 196 patch tokens as nodes. For a core ratio of 10%, around 20 patch tokens are considered as cores, and we choose the top 20 important regions as cores. This partitioning method is inspired by human brain networks [60], where different networks exhibit different core ratios. This architecture allows BNNs to effectively enhance information transmission and communication for integrative processing [5, 19]. To incorporate the Core-Periphery principle into the self-attention mechanism of ViT, we redefined the self-attention operations based on the generated Core-Periphery (CP) graphs, where the patches are regarded as nodes, and the new self-attention relationships are represented by edges in the CP graph. Following this representation paradigm, a complete graph can depict the self-attention of the vanilla ViT. Similarly, the infusion of the Core-Periphery principle into the ViT architecture is achieved by enhancing the complete graph with the generated CP graphs effectively and conveniently. The new self-attention rules can then be redefined: CP graph can be represented by $\mathcal{G} = (\mathcal{V}, \mathcal{E})$, with nodes set $\mathcal{V} = \{\nu_1, ..., \nu_n\}$, edges set $\mathcal{E} \subseteq \{(\nu_i, \nu_j) | \nu_i, \nu_j \in \mathcal{V}\}$, and adjacency matrix $A$. The CP graph guided self-attention for a specific node $i$ at $r$-th layer of ECP-ViT is defined as:

$$x_i^{(r+1)} = \sigma^{(r)}(\{(\frac{q_i^{(r)}(K_j^{(r)})^T}{\sqrt{d_k}})V_j^{(r)}, \forall j \in N(i)\}),$$ (1)

where $\sigma(\cdot)$ is the softmax function, $q_i^{(r)}$ is the query of patches in the $i$-th node in $\mathcal{G}$, $N(i) = \{i | i \vee (i,j) \in \mathcal{E}\}$ are the neighborhood nodes of node $i$, $d_k$ is the dimension of queries and keys, and $K_j^{(r)}$ and $V_j^{(r)}$ are the key and value of patches in node $j$.

In ECP-ViT, each node can contain a single patch or a set of multiple patches. We propose the following patch-assigning pipeline to map the original patches to the nodes of the CP graph. In vanilla ViT with patch size $16 \times 16$, one input image with resolution $224 \times 224$ is divided into 196 patches. When we use a CP graph with $n$ nodes to design the self-attention mechanism, 196 mod $n$ nodes will be assigned $\lfloor 196/n \rfloor + 1$ patches and the remaining $n - (196\ mod\ n)$ nodes will be assigned $\lfloor 196/n \rfloor$ patches. For example, if we use a 5 node CP graph, the 5 nodes will have 40, 39, 39, 39, and 39 patches, respectively; and if we use a 196 nodes CP graph in another case, each node will contain a single patch. Based on the above discussion, the CP graph-guided self-attention that is conducted at the node level can be formulated as:

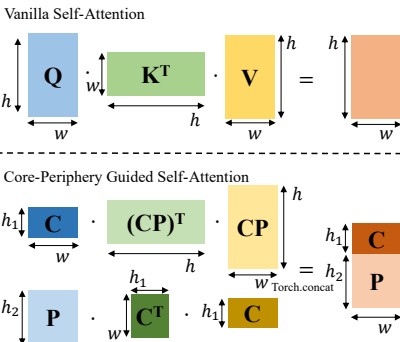

Figure 5: The Core-Periphery Principle guides Self-Attention, where the Query, Key, and Value matrices are partitioned into core (C) and periphery (P) components. The conventional self-attention mechanism is transformed into Core-Periphery (CP) attention through the guidance of Core-Periphery graphs.

$$\text{CPAttention}(Q, K, V) =$$
$$\text{concat}(\text{softmax}(\frac{Q_c K^T}{\sqrt{d_k}})V, \text{softmax}(\frac{Q_p K_c^T}{\sqrt{d_k}})V_c), \quad (2)$$

where queries, keys, and values of all patches are packed into matrices $Q$, $K$, and $V$, respectively, and subscript $c$ and $p$ represent the core parts and periphery parts of the matrices. The graphical illustration of CPAttention is shown in Figure 5. Similar to the multi-head attention in transformers [48], our proposed CP multi-head attention is formulated as:

$$\text{MultiHead}(Q, K, V) = \text{concat}(head_1, ..., head_h)W^o,$$
$$\text{where}\ \ head_i = \text{CPAttention}(QW_i^Q, KW_i^K, VW_i^V), \quad (3)$$

where the parameter matrices $W_i^Q$, $W_i^K$, $W_i^V$ and $W^O$ are the projections. Multi-head attention helps the model to jointly aggregate information from different representation subspaces at various positions. In this work, we apply the CP principle to each representation subspace. Therefore, the self-attention guided by CP graphs in ECP-ViT reduces the computational budgets, while still maintaining the hardware-friendly computation pattern, i.e., converting the traditional self-attention into two small parts of matrix multiplication.

### 3.3 Flexible Data Layout Selection

DNN execution on mobile devices is in a manner of layer-wise computational graph (CG). The CG consists of nodes and edges, each node is an operator such as `MatMul` or `LayerNorm`, and the edge is an indicator to show the direction of the data flow. The main idea behind our optimization for eliminating data transformation is that, in a CG, each operator (such as `MatMul` or `Convolution`) has both producers and consumers. *The producer generates a layout based on the consumer's preferred data layout, resulting in relatively low additional overhead compared to explicitly reorganizing the data.* For instance, if a `MatMul` is followed by a `Transpose` and another `MatMul`, we can make the first `MatMul` directly generate the desired data layout for the second `MatMul`, thus avoiding the need for explicit data reorganization within `Transpose`. Our compiler optimizations consist of three steps: (i) Identify nodes to fuse; (ii) Determine possible data layouts for the key nodes; (iii) Evaluate possible data layouts. We elaborate on them in the following sections.

**Identify nodes to eliminate** starts by classifying the operator in CG into two types: 1) *key nodes* (nodes that perform the actual computation such as `MatMul`, `Softmax`, and `Add` are called *key*

*nodes*; and 2) nodes that only do layout transformation such as `Slice` and `Transpose` are called *transformative nodes*, which are the targets to eliminate. We first use the `transformative node` as a breaking point to partition the graph into a set of sub-graphs. We also apply the fusion rules similar to [37] to identify all fusion opportunities within the sub-graphs. To fully eliminate the data transformation operations, our general strategy is to find a common data layout which works efficiently for both contiguous sub-graphs. Note that, in this step, `transformative nodes` are still kept in the sub-graph because *key nodes'* data layouts are not the same.

**Determine and evaluate possible data layouts** are to find out the best intermediate data layout to eliminate the necessity of introducing additional operators solely for layout transformations. We conduct an exhaustive evaluation of all feasible data format selections, intending to optimize hardware accelerations. It is worth noting that varying data layouts result in distinct access patterns within the mobile GPU, as shown in Figure 6. Notably, noncontinuous data access patterns yield inferior data locality, thereby leading to increased latency. Our *dimension reduction heuristic* is aiming to find the reduction dimension(s) from both ends and group the reducing dimension continuously in the memory, in order to avoid the undesired data accessing pattern. For example, in a `Matmul` operation with $A_{m,k}$ and $B_{k,n}$, the k dimension is our reducing dimension.

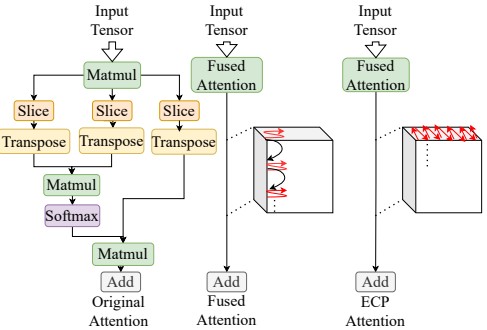

Figure 6: Data Access Pattern Comparisons. On the leftmost, it is the naive attention computational flow graph. In the middle part, it is the strided access pattern without any optimizations, and the rightmost graph is our optimized data access pattern.

Furthermore, specific operators in the subsequent operators may suitable for specific data formats. Take the `LayerNorm` operation as an example, it usually applies the calculation in one dimension only, we generally arrange the target dimension in the width continuously, and group other dimensions into the height dimension. It gives better data locality and GPU utilization. After using our heuristic algorithm to find the best data layout for each *key node*, we are then able to calculate the index transformation from the current node to its child nodes and map the index at the kernel level, which essentially eliminates all the trans-formative operations. Additionally, we store the whole intermediate results within the GPU texture memory to mitigate the time required for data transfer between the CPU and GPU during inter-operator operations. This adjustment is critical, given that data transfer speed between the GPU and CPU remains a persistent bottleneck in mobile device performance.

### 3.4 Compression-compilation co-design in ViT

We evaluate the speed of pruned models by utilizing compiler code generation and on-device latency measurement before performing a time-consuming pruning process. The compiler does not need absolute weight values for code generation and latency measurement, making the process more streamlined. Code generation with the compiler is much faster than Deep Neural Network (DNN) training. Therefore, before actually pruning the models, we use the compiler to evaluate them with different pruning ratios, using synthesized weights instead of real ones. This approach helps us establish a predictive curve that shows how pruning ratios correlate with expected latency. Such a strategy is crucial in determining the optimal pruning ratio that balances model performance and computational efficiency. It allows for a more effective decision on pruning before undertaking the computationally intensive DNN training process.

## 4 Evaluation

In this section, we evaluate the performance of our compiler-assisted framework with our *ECP-ViT* model deployed on mobile devices. We use CP-Level and core ratio interchangeably in the following section.

### 4.1 Setting

**Models and datasets.** The ECP-ViT is implemented based on the ViT architecture [17] and evaluated on 4 different datasets, STL-10 [12], CIFAR-100 [27], TinyImageNet [29], and ImageNet-1K [15]. TinyImageNet is a subset of ImageNet-1k containing 100,000 images distributed across 200 classes. The parameters of ECP-ViT were initialized and fine-tuned from ViT-B/16 trained on ImageNet-21K [28] for TinyImageNet and ImageNet, and used parameters from pre-trained ViT-S/16 for STL-10 and CIFAR-100. We trained the ECP-ViT for 50 epochs with batch size 256 for STL-10, CIFAR-100, and 128 for TinyImageNet and ImageNet-1K, and used AdamW optimizer and cosine learning rate schedule [35] with an initial learning rate of $5e-4$ and minimum of $1e-7$.

**Evaluation environment.** We compare the latency of ECP-ViT on off-the-shelf mobile devices against three state-of-the-art deep learning frameworks: Alibaba MNN [26], Tencent TNN, and Apache TVM [11].

We do not include emerging DNN inference frameworks like TFLite[2] and PyTorch Mobile[25] because they do not support ViT on mobile GPUs yet due to unsupported operators or insufficient resources (e.g., memory capacity). Our evaluation focuses on GPU instead of CPU or NPU for two reasons. Firstly, compared to mobile CPUs, mobile GPUs offer higher computational capacity with better power efficiency. Secondly, compared to mobile NPUs, the NPUs backend is often invoked via a system call provided by the Android Runtime System or specific hardware vendors which does not provide an interface for independent developers to support or optimize certain operators yet. We leave this as a future research direction.

The evaluations are conducted on a Oneplus 11 cell phone, which features a high-end Qualcomm Kryo octa-core CPU and a Qualcomm Adreno 740 GPU with 16 GB of unified memory. To demonstrate the portability of our methods, we also present results from testing on a low-end cell phone - Xiaomi 6 with limited memory and computation capacity, as shown in Figure 7. Xiaomi 6 is equipped with an ARM Octa-core CPU, an Adreno 540 GPU, and 6 GB of unified memory. We use 16-bit floating point precision across all frameworks and models on the mobile GPU. All latency data is collected from running the tests 50 times. However, since the variance is small, we only report averages.

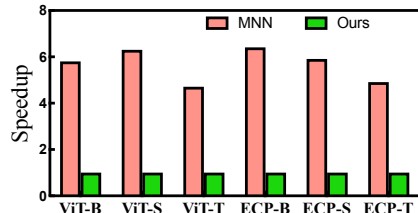

Figure 7: Latency Speedup Comparison Over Low-end Device (Xiao Mi 6). B, S, T short for Base, Small, and Tiny models, respectively.

### 4.2 Accuracy Comparison

The accuracy comparison with other works (with pruning and without pruning) is presented in Table 2. Notably, the accuracy of ECP-ViT is under the best ratio, showcasing its effectiveness. The core-periphery principle guided self-attention of ECP-ViT proves to be competitive with self-attention in a complete graph form. This demonstrates that the message exchange of core-periphery, derived from brain function networks, leverages the message communication in ViT.

We also assess the performance of the proposed ECP-ViT on ImageNet-1K with varying core ratios, and the results are detailed in Table 3. The baseline, denoted as vanilla ViT, featuring self-attention in a complete graph form, is considered to possess a core ratio of $1.0$. Notably, for the ViT-base model scale, our ECP-ViT-base, integrating the core-periphery principle, demonstrates superior performance compared to ViT-base across a spectrum of core ratios $(0.7, 0.8, \text{and } 0.9)$. ECP-ViT surpasses the baseline by 0.73% under a ratio of 0.9, suggesting that the sparsity of self-attention could enhance performance, with the core-periphery principle guiding self-attention proving to be an effective means of achieving this sparsity.

It's worth noting that our baseline is fine-tuned by us and already stands out compared to that reported in other works [65, 17, 47, 14]. Even for smaller model scales, such as ECP-ViT-small and ECP-ViT-tiny, the drop in accuracy is minimal when compared to vanilla ViT. Additionally, we conducted a comparative analysis of ECP-ViT with other competitive models on TinyImageNet, as depicted in Table 4. The results highlight the advantages of ECP-ViT in terms of prediction accuracy.

Table 2: Comparisons of results on the ImageNet dataset. All reported model results are based on pre-trained weights from ImageNet-21K. The accuracy, parameters, and MACs of ECP-ViT are under the best core ratio.

| Model | Top-1 (%) | # Params. | # MACs |
|---|---|---|---|
| LTMP [8] | 75.4 | 5.7M | 1.5G |
| PVT-Tiny [50] | 75.1 | 13.2M | 1.97G |
| ECP-ViT-Tiny | 75.8 | 5.83M | 1.1G |
| PVT-Small [50] | 79.8 | 24.5M | 3.89G |
| ViT-S [17] | 81.1 | 22M | 4.62G |
| T2T-ViT$_t$-14 [63] | 80.7 | 22M | 4.8G |
| ECP-ViT-Small | 80.9 | 21.7M | 4.3G |
| ViT-Base/16 [17] | 83.9 | 86.6M | 17.6G |
| PVT-Large [50] | 83.8 | 82.0M | 11.84G |
| TNT-B [20] | 84.1 | 66.0M | 14.16G |
| DeiT-Base/16 [45] | 84.2 | 86.6M | 17.76G |
| ECP-ViT-Base | 84.6 | 86.5M | 16.96G |

Table 3: Performance evaluation of ECP-ViT on ImageNet under different core ratios. We fine-tune the ECP-ViT using the pre-trained weights on ImageNet-21K. Top-1 accuracy is reported and shown in percentage. T, S, and B represents ECP-ViT-T, ECP-ViT-S, ECP-ViT-B, respectively.

| Ratio | 0.1 | 0.2 | 0.3 | 0.4 | 0.5 | 0.6 | 0.7 | 0.8 | 0.9 | 1.0 (base) |
|---|---|---|---|---|---|---|---|---|---|---|
| T/16 | 65.50 | 69.14 | 69.39 | 69.75 | 71.38 | 73.13 | 74.55 | 75.48 | 75.84 | 75.90 |
| S/16 | 73.24 | 74.42 | 74.98 | 76.32 | 77.82 | 78.78 | 79.82 | 80.34 | 80.88 | 81.12 |
| B/16 | 76.15 | 77.98 | 79.40 | 80.51 | 81.80 | 83.22 | 84.03 | 84.51 | 84.62 | 83.89 |

Table 4: Comparison between ECP-ViT with other ViT variants on TinyImageNet. Top-1 Accuracy is shown in percentage. The best result of ECP-ViT under different core ratios is selected.

| Model | Top1 Acc. | Params. | Image Res. |
|---|---|---|---|
| ViT-B/16 [17] | 89.16 | 86.55M | 224 × 224 |
| DeiT-B/16 [46] | 87.29 | 87.34M | 224 × 224 |
| BeiT-B/16 [4] | 88.64 | 86.53M | 224 × 224 |
| ConViT-B/16 [14] | 90.52 | 86.54M | 224 × 224 |
| ECP-ViT-B/16 | **90.55** | 86.50M | 224 × 224 |

Table 5: Comparisons between ECP-ViT and vanilla ViT on STL10 and CIFAR100. The performance evaluation is under the best core ratio. Top-1 accuracy is reported and shown in percentage.

| Model | STL-10 | CIFAR-100 |
|---|---|---|
| ViT-S/16 [17] | 96.36 | 89.51 |
| ECP-ViT-S/16 | 98.18 (+1.82) | 89.56 (+0.05) |

Furthermore, we extend our evaluation to STL10 and CIFAR-100, with detailed results provided in Table 5.

## 4.3 End-to-end Latency and Memory Comparison

Table 6 compares peak memory, latency, and cache miss rates between ViT-Base and ECP-ViT. Table 7 presents a comparison of latency among ECP-ViT, MNN, TNN, and TVM for vanilla ViT models. As the models retain their dense structure after CP pruning, they can still be executed on other frameworks. '-' means the model is not supported on the framework, due to lack of operator implementation or limited memory/computation resources. As shown in Table 7, compared to other state-of-the-art frameworks, ECP-ViT achieves an average speedup ranging from $4.8\times$ to $5.3\times$ for vanilla ViTs. This is because our compiler optimizations, such as data transformation elimination and operator fusion, help significantly reduce memory pressure and bandwidth demands. Moreover, our compiler optimizations for ECP-ViTs leverage the decreased computational complexity and yield a speedup ranging from $10.6\times$ to $16.1\times$ compared to vanilla ViT within our framework. Additionally, ECP-ViT outperforms other frameworks with a speedup ranging from $4.6\times$ to $26.9\times$. TVM exhibits

even higher latency for ECP-ViT due to the absence of systematic optimizations. The extra data transformation resulting from CP pruning leads to heightened demands on memory bandwidth. Our framework achieves real-time performance for small and tiny variants of ECP-ViT, meeting the requirement for real-time execution at 30 frames per second.

Table 6: Comparison of Peak Memory, Latency, and Miss Rates for ViT-Base and ECP-ViT.

| Model | Peak Memory (MB) | Latency (ms) | Cache Miss Rate (%) | | |
| --- | --- | --- | --- | --- | --- |
| | | | L1 | L2 | L3 |
| ViT-Base[48] | 454 | 421.25 | 0.77 | 5.94 | 15.12 |
| ECP-ViT | 403 | 99.84 | 0.66 | 5.38 | 15.05 |

Table 7: Latency comparison of 4 end-to-end frameworks on vanilla ViTs and CP-enabled ViTs using the GPU on Oneplus 11. We use CP level of 80% for all 3 variants (base, small and tiny). '-' means the models is not supported on the framework.

| Model | TNN (ms) | TVM (ms) | MNN (ms) | Ours (ms) | Speedup (avg) |
| --- | --- | --- | --- | --- | --- |
| VIT-tiny | 300.0 | 54.7 | 115.3 | **17.6** | **4.8** |
| VIT-small | - | 176.2 | 191.9 | **37.7** | **4.9** |
| VIT-base | - | 780.3 | 610.3 | **131.9** | **5.3** |
| ECP-VIT-tiny | - | 380.1 | 110.1 | **15.2** | **16.1** |
| ECP-VIT-small | - | 837.6 | 157.9 | **31.1** | **16.0** |
| ECP-VIT-base | - | 2033 | 563.2 | **122.8** | **10.6** |

## 5 Conclusion

This paper introduces ECP-ViT, a framework that enhances the deployment of Vision Transformer (ViT) models on mobile devices for real-time AI applications. Our approach includes a hardware-friendly pruning method inspired by the brain network and a set of compiler optimizations to eliminate data transformation bottlenecks. The results show that ECP-ViT not only reduces computational size but also improves prediction accuracy. It achieves up to 26.9x speedup compared to other state-of-the-art frameworks, enabling real-time performance on off-the-shelf mobile devices.

## 6 Impact Statement

This paper aims to advance the real-time implementation of brain-inspired AI on mobile devices. The integration of the brain-inspired core-periphery principle contributes to reducing the computation budget and enhancing prediction accuracy. Additionally, the optimization of hard-level data layout could significantly improve inference speed on mobile devices. This work has the potential to boost the development of brain-inspired AI on mobile devices.

## Acknowledgment

We would like to express our gratitude to all those who contributed to this work, with special thanks to the constructive comments from the anonymous reviewers. This work was supported in part by the National Science Foundation (NSF) under the awards of CCF-2428108, OAC-2403090 . Any errors and opinions are not those of the NSF and are attributable solely to the author(s).

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
