# OpenReview forum: "Real-time Core-Periphery Guided ViT with Smart Data Layout Selection on Mobile Devices"
_NeurIPS.cc/2024/Conference — NeurIPS 2024 poster_

### Official Review · Reviewer_wumZ · 2024-07-04

**Soundness:** 3
**Presentation:** 3
**Contribution:** 2
**Rating:** 4
**Confidence:** 3

**Summary:**

• This paper proposes ECP-ViT, which optimizes the ViT model by introducing the core-periphery (CP) principle.
• The Core-Periphery Principle Guided self-attention mechanisms successfully reduce memory bandwidth by eliminating data transformation.
• By applying pruning and removing data transformation, the optimization, which considers both software-level design (algorithm) and hardware-level design, achieves real-time performance on a mobile GPU (Snapdragon 8 Gen 2 SoC) with an average speedup of 4.8 times.

**Strengths:**

• The performance generality of the proposed ECP-ViT is demonstrated across various datasets (STL-10, CIFAR-100, TinyImageNet, and ImageNet) and hardware environments (OnePlus 11, Xiaomi 6).
• The justification for the proposed Core-Periphery Guided Self-Attention is well-explained, including background information, and effectively illustrated with figures and equations.
• ECP-ViT achieves speed improvements by completely eliminating data transformation, which impacts memory bandwidth and causes network overhead.
• (Table 8) The compiler speed is significantly improved compared to the MNN and TVM frameworks.
• (Table 9) The overhead of data layout transformation and computation for the mobile GPU is analyzed in detail.

**Weaknesses:**

• The comparative experiments lack consistency.
• Overall, the performance improvements are marginal compared to previous research.
• The implementation uses Fixed Point 16-bit, but most current mobile environments typically utilize 8-bit or higher quantization, and there are no experiments addressing this. Data compression through quantization is essential for mobile environments.
• More fair comparisons could be made if experiments were conducted in specific NPU environments.
• Although a compiler for mobile GPUs is proposed, detailed information about the compiler itself is lacking.
    o (Figure 4) There is insufficient explanation on how the specific allocation of the core/periphery nodes is determined, and there is no criterion for specifying core nodes.
    o The detailed operation of the proposed dimension reduction heuristic algorithm is not explained. More detailed descriptions and algorithms are needed to optimize data layout.

**Questions:**

• (Table 1, Table 9) Are there experimental results on latency for layout transformation and computation in other frameworks such as TFLite and Pytorch-Mobile?
    o The experimental setup in Table 9 indicates a batch size of 1 to 18. Is this an average value? Do the experimental results for latency ratios align with different batch sizes?
• (Table 3) Are there any comparative experimental results for Pytorch-Mobile?
• (Figure 6) Are there experimental results on the computation complexity and memory usage of the model according to the core ratio? Are there results consistent with those presented in Table 1?
• (Table 6) Were the experiments for TNN, TVM, and MNN conducted on NPUs while those for Ours were conducted on a mobile GPU? Are there experimental results for Ours on NPUs?
• Are there experimental results for tasks such as Object Detection or Instance Segmentation, where real-time inference is crucial?

---

> ### Author Rebuttal · Authors · 2024-08-06
>
> ### Q1 (Table 1, Table 9) Are there experimental results on latency for layout transformation and computation in other frameworks such as TFLite and Pytorch-Mobile?
>
> **Response:**
> | Model                   | Implicit Transformation (ms) | Explicit Transformation (ms) | Computation (ms) | Latency (ms) |
> |-------------------------|------------------------------|------------------------------|------------------|--------------|
> | ECP-ViT (our framework) | 0                            | 0                            | 99.83            | 99.83        |
> | ViT-Base                | 198.93                       | 0.92                         | 134.52           | 334.37       |
> | DeiT-Base               | 216.26                       | 5.294                        | 136.09           | 357.64       |
> *Device – Oneplus 12 Snapdragon 8 Gen 3 SoC
>
> *Platform – TFLite
>
> ### Q2 The experimental setup in Table 9 indicates a batch size of 1 to 18. Is this an average value? Do the experimental results for latency ratios align with different batch sizes?
>
> **Response:**
>
> Thank you for your question regarding the experimental setup in Table 9. We actually tested all batch sizes from 1 to 18 to ensure comprehensive evaluation. However, we observed that there were no major differences in latency across these different batch sizes. As a result, for simplicity and clarity in our presentation, we reported the results for batch size 1. The latency ratios remained consistent regardless of the batch size, ensuring that our conclusions are robust across varying batch sizes.
>
> ### Q3 (Table 3) Are there any comparative experimental results for Pytorch-Mobile?
>
> **Response:**
>
> Thank you for your question regarding comparative experimental results for Pytorch-Mobile in Table 3. Currently, Pytorch-Mobile does not support ViT model on mobile GPU. Therefore, we use TensorFlow Lite (TFLite) for our experiments instead and conducted all tests using OpenCL on GPUs. This approach allows us to provide a fair and comprehensive comparison of performance across different frameworks that support Vision Transformer models on mobile GPUs. Please refer to Q7 for the reason of choosing GPU instead of NPU.
>
> | Model      | Framework | Latency (ms) |
> |------------|-----------|--------------|
> | ECP-ViT    | Ours      | 99.83        |
> | ViT-Base   | TFLite    | 334.37       |
> | DeiT-Base  | TFLite    | 357.64       |
>
> *Device – Oneplus 12 Snapdragon 8 Gen 3 SoC
>
> *Results are averaged by running 10 rounds and 5 warm_ups
>
> ### Q4 (Figure 6) Are there experimental results on the computation complexity and memory usage of the model according to the core ratio? Are there results consistent with those presented in Table 1?
>
> **Response:**
>
> | Model      | Peak Memory In core ratio 0.6 | Peak Memory In core ratio 0.7 | Peak Memory In core ratio 0.8 | Peak Memory In core ratio 0.9 |
> |------------|-------------------------------|-------------------------------|-------------------------------|-------------------------------|
> | ECP-ViT    | 361 MB                        | 378 MB                        | 403 MB                        | 420 MB                        |
>
> These results are consistent with those presented in Table 1. After pruning, we separate computation into core-to-core and core-to-periphery nodes. The small size of core nodes results in intermediate results being similar across different core ratios. Additionally, our layout elimination technique stores intermediate results on the GPU and reuses finished buffers, saving more memory compared to other frameworks. This efficient handling keeps memory usage stable across different core ratios, aligning with the results in Table 1 and demonstrating the effectiveness of our ECP-ViT model in managing computation complexity and memory usage.
>
> ### Q5 Are there experimental results for tasks such as Object Detection or Instance Segmentation, where real-time inference is crucial?
>
> **Response:**
> |                 | DETR [1] | SegFormer [2] |
> |-----------------|----------|---------------|
> | Vanilla         | 42.0     | 83.8          |
> | w/ Our Method   |  42.6         |   84.2            |
>
> [1] End-to-end object detection with transformers. In ECCV 2020.
>
> [2] SegFormer: Simple and efficient design for semantic segmentation with transformers. In NeurIPS 2021.
>
> ### Q6 Why not use quantization?
>
> **Response:**
>
> Our paper focuses on layout elimination techniques to reduce memory burden without sacrificing accuracy. Our main contribution lies in optimizing data layout and computation patterns, eliminating data transformation overhead, and enhancing performance on mobile devices. It’s important to note that quantization and our optimization techniques are orthogonal and can be applied independently or together for cumulative benefits. We plan to extend our research to include experiments with 8-bit quantization, aligning with current mobile environment practices.
>
> ### Q7  Why Not using NPU?
>
> **Response:**
>
> All the experiments were conducted on mobile GPUs because most frameworks like MNN or TNN do not support NPUs. We did not conduct experiments using NPUs because mobile NPUs, while faster than GPUs, lack low-level programmable interfaces for individual developers [1]. TFLite uses an NPU backend via system calls provided by the Android Runtime System, which do not provide an interface for independent developers to support or optimize certain operators [1].
>
> We have to put the comparison table in the PDF due to space limit. While the NPU theoretically outperforms the GPU, our GPU implementation of ECP-ViT runs at almost the same speed as TFLite’s NPU, demonstrating our framework’s efficiency in leveraging GPU capabilities effectively.
>
> Qualcomm AI hub devices do not support Oneplus 12 yet, for TFLite with NPU we are using remote device Samsung Galaxy S24+ provided by Qualcomm AI hubs [1]
>
> [1] The Qualcomm® AI Hub Models https://github.com/quic/ai-hub-models?tab=readme-ov-file

---

### Official Review · Reviewer_PnBL · 2024-07-06

**Soundness:** 4
**Presentation:** 3
**Contribution:** 3
**Rating:** 6
**Confidence:** 5

**Summary:**

This paper introduces ECP-ViT, a framework that accelerates Vision Transformers on mobile devices using a core-peripheral guided self-attention mechanism. This approach reduces computational demands and achieves up to 16.1x speedup on a OnePlus 11 GPU, enabling efficient real-time deployment of ViT models while maintaining high accuracy.

**Strengths:**

- The compiler and model architecture were co-optimized, significantly improving computational efficiency on actual devices.
- Based on the Brain Neural Networks, this paper cleverly distinguishes between important and less important parts of the attention mechanism.
- The slice and transpose reshape operations are eliminated through the hardware-level design.

**Weaknesses:**

- Overall, I think this paper is above the acceptance bar. However, the comparison (or discussion) between ECP-ViT and other lightweight ViT methods is not comprehensive enough, for example, NAS for lightweight ViT [1][2], and advanced token optimization method [3].
- Showing the results on ImageNet seems like a reasonable choice, but insufficient to show that the method can be generalized to other domains, such as detection/segmentation. One possible way to do this is to directly transfer the weights for another downstream task and provide some comparison. By the way, the performance improvements in Tab.11 seem a bit marginal compared to the main results.

[1] Elasticvit: Conflict-aware supernet training for deploying fast vision transformer on diverse mobile devices, ICCV 2023

[2] Nasvit: Neural architecture search for efficient vision transformers with gradient conflict-aware supernet training, ICLR 2022

[3] Diffrate: Differentiable compression rate for efficient vision transformers, ICCV 2023

**Questions:**

How does the core rate be selected for different devices and tasks? Would making it a sample-aware parameter further enhance the results?

**Limitations:**

Please refer to the weakness.

---

> ### Author Rebuttal · Authors · 2024-08-06
>
> ### Q1 More comparison between other lightweight ViT methods.
>
> [1] Elasticvit: Conflict-aware supernet training for deploying fast vision transformer on diverse mobile devices, ICCV 2023
> [2] Nasvit: Neural architecture search for efficient vision transformers with gradient conflict-aware supernet training, ICLR 2022
> [3] Diffrate: Differentiable compression rate for efficient vision transformers, ICCV 2023
>
> **Response:**
> | Model         | Top1 Acc on ImageNet-1k | Computation size (GFLOPs) |
> |---------------|-------------------------|--------|
> | ECP-ViT       | 84.6                    | 10.1   |
> | ElasticViT-L3 | 80.0                    | 0.86   |
> | NasViT-A5     | 81.8                    | 0.76   |
> | DiffRate-Base | 81.5                    | 11.5   |
>
> As shown in the table, ECP-ViT achieves the highest top-1 accuracy on ImageNet-1k at 84.6%, demonstrating superior performance compared to other lightweight ViT methods. We will add the comparison and the speed comparison in the revision
>
> ### Q2 Show results on other domains.
>
> **Response:**
>
> Thank you for the suggestion. We have further applied the core-periphery guided ViT to detection and segmentation tasks, specifically using DETR on the COCO dataset and SegFormer on the Cityscapes dataset.
>
> |                 | DETR [1] | SegFormer [2] |
> |-----------------|----------|---------------|
> | Vanilla         | 42.0     | 83.8          |
> | w/ Our Method   |    42.6      |       84.2        |
>
> [1] End-to-end object detection with transformers. In ECCV 2020.
>
> [2] SegFormer: Simple and efficient design for semantic segmentation with transformers. In NeurIPS 2021.
> A major contribution of our work emphasizes real-time performance on mobile devices. Our method reduces parameters and latency while maintaining or even improving model performance on devices. Although the performance improvements in Table 11 may seem marginal, they demonstrate that our method achieves a balance between efficiency and effectiveness.
>
>
> ### Q3 How does the core rate be selected for different devices and tasks? Would making it a sample-aware parameter further enhance the results?
>
> **Response:**
>
> Thank you for your question regarding the selection of the core ratio for different devices and tasks.
> Since our core-periphery principle guided ViT is inspired by brain functional networks, where different networks responsible for various tasks (such as vision and movement) have different core ratios, we have adopted a similar approach that we experiment with different core ratios to find the optimal ratio for specific datasets or tasks. Your suggestion to make it a sample-aware parameter is indeed interesting and a great idea for further exploration.
> According to Figure 8 and Figure 6, we employ different core ratios to run benchmarks and obtain a performance line in terms of accuracy. We then test the latency for these different core ratios. By evaluating both accuracy and latency metrics, we can identify the optimal core ratio that balances performance and efficiency. This systematic approach ensures that the selected core ratio is well-suited for the specific device and task, providing optimal results. Additionally, making the core ratio a sample-aware parameter could potentially further enhance results by allowing dynamic adjustment based on the specific characteristics of the data, thereby improving both accuracy and efficiency.

---

### Official Review · Reviewer_MoNf · 2024-07-12

**Soundness:** 2
**Presentation:** 2
**Contribution:** 3
**Rating:** 3
**Confidence:** 3

**Summary:**

This paper presents ECP-ViT, a real-time framework for deploying Vision Transformers (ViTs) on mobile devices. Inspired by the brain's core-periphery principle, this method guides self-attention in ViTs to reduce computational demands and eliminate data transformation operations. ECP-ViT integrates algorithm-system co-optimizations, achieving a speedup of 4.6× to 26.9× on mobile GPUs across various datasets while maintaining high accuracy.

**Strengths:**

1. The motivation data in Table 1 is interesting.

2. This paper targets an interesting problem: eliminating the expensive transform operation in ViTs.

**Weaknesses:**

1. The paper writing can be improved.

2. Some figures and their elaborations are not clear enough.

3. The rationale behind the design is not well-explained.

Please see my comments.

**Questions:**

1. In line 67 of the introduction, it mentions support for a pruning scheme, which is confusing because there is no prior content about pruning before this line.

2. In the background section, it states that the token pruning method introduces additional Reshape and Transpose operations to the feature map, leading to reduced benefits from reduced computation costs. However, in many token pruning methods [1] [2], pruning is applied to certain specific layers, so the overhead of Transpose and Reshape is not that large. Moreover, token pruning quadratically reduces computation cost, which is highly effective.

3. In Section 3.1, it says it classifies data transformation into two categories, Explicit and Implicit, and Explicit transformations are denoted as red color operators in round boxes. But there are several colors in Figure 2, and some colors are quite similar, making it hard to distinguish between Explicit and other transformations. Also, it seems that there is no indication of the implicit transformations in the figure.

4. It says the CP-guided self-attention design is shown in Figure 4. But Figure 4 seems to depict the ratio of core nodes and the number of edges. This is confusing as it does not illustrate any design component or workflow.

5. The rationale behind the CP-guided self-attention design is not detailed enough. CP-guided self-attention is an efficient self-attention mechanism that reduces the number of self-attention operations, but why is it better than other efficient self-attention mechanisms such as Swin-Transformer and Criss-Cross Attention? Why do we adopt this CP-guided self-attention? How do you ensure that the information exchange through only core nodes is sufficient?
&nbsp;

[1] Liang, Youwei, et al. "Not all patches are what you need: Expediting vision transformers via token reorganizations." arXiv preprint arXiv:2202.07800 (2022).

[2] Kong, Zhenglun, et al. "Peeling the onion: Hierarchical reduction of data redundancy for efficient vision transformer training." Proceedings of the AAAI Conference on Artificial Intelligence. Vol. 37. No. 7. 2023.

**Limitations:**

Please see my comments.

---

> ### Author Rebuttal · Authors · 2024-08-06
>
> ### Q1 Confusions in line 67 of the introduction.
>
> **Response:**
>
> Thank you for the suggestion. We will revise the introduction to provide a brief context about pruning before mentioning our support for it. Specifically, we will add a paragraph to explain the concept and relevance of pruning in the context of our work.
>
>
> ### Q2  Token pruning methods add some overhead with Reshape and Transpose operations but are still highly effective in reducing computation costs by being applied to specific layers.
>
> [1] Liang, Youwei, et al. "Not all patches are what you need: Expediting vision transformers via token reorganizations." arXiv preprint arXiv:2202.07800 (2022).
>
> [2] Kong, Zhenglun, et al. "Peeling the onion: Hierarchical reduction of data redundancy for efficient vision transformer training." Proceedings of the AAAI Conference on Artificial Intelligence. Vol. 37. No. 7. 2023.
>
> **Response:**
>
> Thank you for your feedback regarding the token pruning methods. Token pruning methods [1][2] can result in sparse data patterns that implicitly require more data transformations. Implicit data transformations occur when the underlying data layout needs to be reorganized to optimize performance for specific operations, leading to irregular data access patterns and increased memory bandwidth demands. This can be particularly unfriendly for mobile devices, which have limited memory bandwidth.
> In ECP-ViT, we address this issue by grouping important patches together and pruning at the node level. This approach ensures that even though the model is pruned, each sub-matrix remains dense, allowing for more efficient computation. By maintaining dense computation patterns, ECP-ViT minimizes the need for implicit data transformations, thereby enhancing performance and efficiency on mobile devices. This makes ECP-ViT more suitable for mobile deployment compared to methods that result in sparse data patterns and increased implicit data transformations.
>
>
>
>
>
> ### Q3  In Section 3.1, it states that data transformations are classified into Explicit (red, round boxes) and Implicit categories, but Figure 2 uses several similar colors making it difficult to distinguish Explicit transformations, and lacks indication of Implicit transformations.
>
> **Response:**
>
> Thank you for your observation regarding the classification of data transformations in Section 3.1 and their representation in Figure 2. We will update the hardware design in the revision. In the updated figure, every computational operator will have its preferred data layout, and between each computational operator, there will potentially be an implicit reshape or transpose. These implicit reshapes occur because different computational operators often require their input data in specific formats or layouts for optimal performance. When the output layout of one operator does not match the required input layout of the next, an automatic reshape or transpose is needed to ensure compatibility and efficiency. In our framework, by using smart layouts, we will eliminate the implicit reshapes. Additionally, we will ensure that the colors used to denote different types of operations are distinct and easily distinguishable.
>
>
> ### Q4 Confusions in Figure 4 (CP-Guided self-attention)
>
> **Response**
>
> Thanks for pointing this out. We will revise the sentence to clarify that Figure 4 shows examples of core-periphery graphs with different core ratios. Specifically, Figure 4 illustrates the selection of various CP graphs referenced in Figure 2(a). The workflow involves CP graph generation, CP-guided self-attention, and CP-guided QKV multiplication, corresponding to Figure 2(a), Figure 2(b1), and Figure 2(b2).
>
>
> ### Q5 More details are needed for CP-guided self-attention design. Why is it better than Swin and Criss-Cross Attention? Only core nodes are sufficient?
>
>
> **Response:**
>
> Thank you for your question regarding the rationale behind the CP-guided self-attention design and its advantages over other efficient self-attention mechanisms such as Swin-Transformer and Criss-Cross Attention.
> From a hardware perspective, CP-guided self-attention can group core patches together and periphery patches together, which increases data locality. This grouping improves the efficiency of data access and reduces the need for frequent data movement across memory, thereby enhancing computational efficiency. While this design involves more implicit and explicit reshape and transpose operations, our framework successfully eliminates both, making it highly suitable for CP-guided self-attention.
> In contrast, mechanisms like Swin-Transformer and Criss-Cross Attention do not inherently focus on optimizing data locality and may still involve significant overhead from data transformations. By reducing these overheads, CP-guided self-attention becomes more efficient and effective, particularly on hardware with limited memory bandwidth like mobile devices. Additionally, the core-periphery principle ensures that information exchange through core nodes is sufficient by maintaining connections among core nodes while selectively connecting periphery nodes, balancing comprehensive context capture with reduced computational complexity. This makes CP-guided self-attention a better fit for our optimization framework compared to other mechanisms.
> Furthermore, our method is actually a pruning mechanism that can also be applied to Swim-Transformer or Criss-Cross Attention. This flexibility allows for the benefits of our approach to be realized in various self-attention-based transformer models, enhancing their efficiency and performance across different applications and hardware configurations.

---

### Official Review · Reviewer_r1Lw · 2024-07-13

**Soundness:** 3
**Presentation:** 2
**Contribution:** 3
**Rating:** 7
**Confidence:** 3

**Summary:**

This paper introduces ECP-ViT, a framework designed to improve the performance of vision transformers (ViTs) on mobile devices. The authors observed the intensive and irregular memory access involved in the data transformation for self-attention layers, which significantly slows down transformers compared to traditional CNNs. To address this, they propose a hardware-friendly self-attention pruning technique motivated by the core-periphery structure in brain networks, thereby reducing the computational and memory access burdens. ECP-ViT also incorporates compiler optimizations to fuse and eliminate transformation operators to further boost the performance. These combined optimizations enable the work to speedup in ViT inference on mobile GPUs by 4.6x to 26.9x without sacrificing the inference accuracy.

**Strengths:**

+ The problem is well-motivated with clear benchmarks in terms of accuracy, MACs, and latency comparisons across various ViTs.
+ The core-periphery concept borrowed from brain networks for self-attention is interesting.
+ The proposed compiler optimization effectively eliminates unnecessary data transformation operators, leading to significant performance improvements.
+ The evaluation is conducted on real-world mobile phones, and includes both high-end and low-end devices.

**Weaknesses:**

- The evaluation section could be improved:
    - It would be beneficial to discuss the memory access pattern differences before and after applying ECP to showcase its effectiveness in improving memory efficiency.
    - The power and energy consumption evaluation should also be included, given the limited battery capacity of mobile devices.
- There are several presentation issues; for example, the space between Table 3 and Table 4 is too dense, and the font size seems to change abruptly starting from the “Evaluation environment” in Section 4. Please ensure consistent font sizes throughout the text.

**Questions:**

1. Can you provide a detailed comparison of memory behavior before and after applying the ECP technique?
2. Can you show the power/energy improvement achieved by the proposed optimizations?

**Limitations:**

The authors have mentioned the plan to evaluate other model architectures in future work, which addresses some limitations. However, it would be beneficial to include additional evaluations such as memory studies, power/energy consumption analysis, and resource utilization metrics.

---

> ### Author Rebuttal · Authors · 2024-08-06
>
> ### Q1  Need to provide a detailed comparison of memory behavior before and after applying the ECP technique.
>
> **Response:**
>
> Thank you for your question regarding the memory access pattern differences before and after applying ECP-ViT. Our pruning technique ensures that computations remain dense, maintaining a memory access pattern similar to the original model. This consistency in memory access patterns, combined with fewer computations, results in faster performance.
> (We will add a fig to illustrate the memory access in the revision) If we just use naive fusion, the access pattern of the tensors between operators will be strided. This strided access pattern is determined by the data transforming operators like slice and transpose. By using our smart layout design, we can map the indices of tensors as desired, and the access pattern will be continuous. Compared to the strided access pattern, our access pattern can utilize the data locality and reduce the cache miss, which shows tremendous reduction in latency.
>
> | Model     | Peak Memory w/ ECP (MB) | Peak Memory w/o ECP (MB) | Latency w/ ECP (ms) | Latency w/o ECP (ms) |
> |-----------|-------------------------|--------------------------|---------------------|----------------------|
> | ViT-Base  | 403                     | 454                      | 99.84               | 421.25               |
>
> *Device – Oneplus 12 Snapdragon 8 Gen 3 SoC
>
> *Results are averaged by running 10 rounds and 5 warm-ups
>
> | Model     | L1 Cache Miss Rate | L2 Cache Miss Rate | L3 Cache Miss Rate |
> |-----------|--------------------|--------------------|--------------------|
> | ViT       | 0.77%              | 5.94%              | 15.12%             |
> | ECP-ViT   | 0.66%              | 5.38%              | 15.05%             |
>
> After ECP pruning, each part of our computation is still dense. Combining with our layout selections, we are able to increase data locality and reduce cache misses in all levels.
>
>
>
>
>
>
> ### Q2 Need to show the power/energy improvement achieved by the proposed optimizations
>
> **Response:**
>
> Testing device: OnePlus 12, Snapdragon 8 Gen 3 SoC
>
> All results are collected by running 10 rounds and pick the peak power usage
>
> | Model            | Power w/o ECP | Power w/ ECP |
> |------------------|---------------|--------------|
> | ViT-Base         | 2.46 W        | 1.05 W       |
> | Swin-Tiny        | 0.87 W        | 0.78 W       |
> | MetaFormer-Base  | 2.74 W        | 1.31 W       |
>
>
> ### Q3 Table 3, 4 and section 4 font issues
>
> **Response:**
>
> Thanks for the suggestions. We will revise it in the revision.

---

> > ### Comment · Reviewer_r1Lw · 2024-08-14
> >
> > Thanks for the additional experiments and explanations. I will keep my score.

---

### Official Review · Reviewer_gvXM · 2024-07-15

**Soundness:** 3
**Presentation:** 3
**Contribution:** 2
**Rating:** 5
**Confidence:** 3

**Summary:**

This work proposes a ViT accelerating framework, ECP-ViT, to deploy ViT models on smartphones. This framework consists of two parts: 1) Core-Periphery Guided Self-Attention (reducing the computational and bandwidth cost of ViT) and 2) Data Layout Selection based on compiler optimizations (removing the time-consuming data transformation). Specifically, Core-Periphery Guided Self-Attention partitions the tokens in K, Q, and V matrices into core and periphery components. Tokens in the core component exchanged messages with all tokens, and tokens in the periphery component only exchanged messages with tokens in the core component. To fully eliminate the data transformation operations, this work attempts to find a common data layout that works efficiently for both contiguous operators. But the search space is large. They propose the dimension reduction heuristic algorithm to shrink the search space of the data layouts. ECP-ViT obtains competitive results on STL-10, CIFAR100, TinyImageNet, and ImageNet, achieving lower smartphone latency.

**Strengths:**

- The Core-Periphery Guided Self-Attention, which is inspired by Brain Neural Networks, is interesting.
- The Data Layout Selection based on compiler optimizations can effectively speed up ViTs on Smartphones, as demonstrated by comprehensive experimental validation.

**Weaknesses:**

- Some details need to be elaborated on. The description of the dimension reduction heuristic needs to be more detailed, and it is encouraged to provide its pseudocode.
- There seems to be a discrepancy in Equation 1. The conventional form of self-attention is softmax(qV)K, while in ECP-ViT, it appears to be softmax(qVK). This discrepancy should be carefully reviewed and corrected.

**Questions:**

1. How do you partition the KQV matrices into core and peripheral components? Are there any partitioning criteria? And why do you use this partitioning method?
2. MobileViT [1] is a lightweight, low-latency network for mobile vision tasks. What significant advantages does ECP-ViT offer over MobileViT that we should consider?

[1] Mehta, Sachin, and Mohammad Rastegari. "Mobilevit: light-weight, general-purpose, and mobile-friendly vision transformer." arXiv preprint arXiv:2110.02178 (2021).

**Limitations:**

Not applicable.

---

> ### Author Rebuttal · Authors · 2024-08-06
>
> ### Q1 Explain dimension reduction heuristic
>
> **Response:**
>
> Thanks for pointing this out. Below is the detailed pseudo code for the heuristic. The score is collected by running mini benchmarks, and the different layouts are defined by the frameworks. The process begins by identifying key nodes in the computational graph that perform actual computation (Step 1). For each key node, possible data layouts are determined (Step 2), taking into account the various layout options provided by the frameworks. Each possible layout is then evaluated by calculating a score based on data locality and GPU utilization using mini benchmarks (Step 2.1). The layout with the best score is selected and assigned to the key node (Step 3). Finally, the computational graph is updated with the new layouts for all key nodes to ensure that the entire graph benefits from the optimized layouts, improving overall performance (Step 4).
>
>
> ### Algorithm 1: Dimension Reduction Heuristic Algorithm
>
> **Require:** Computational graph G
>
> **Ensure:** Optimized computational graph G'
>
> **Step 1: Find all the key nodes**
> - Identify key nodes K ← identify_key_nodes(G)
> - For each node n ∈ K do:
>     - **Step 2: Determine layouts for key nodes by Algo 3**
>     - P ← determine_possible_layouts(n)
>     - best_layout ← None
>     - best_score ← ∞
>     - For each layout l ∈ P do:
>         - **Step 2.1: Calculate scores by mini-benchmarks**
>         - score ← calculate_data_locality_score(l, n) + calculate_gpu_utilization_score(l, n)
>         - If score < best_score then:
>             - best_score ← score
>             - best_layout ← l
>     - End for
>     - **Step 3: Assign best layout for the key node**
>     - n.data_layout ← best_layout
> - End for
> - For each node n ∈ K do:
>     - **Step 4: Infer layout for all nodes**
>     - G.update_node_layout(n)
> - End for
> - Return G
>
>
> ### Q2  A discrepancy in Equation 1.
>
> **Response:**
>
> Thank you for pointing this out. We will correct it in Eq.1 and the corresponding figures. As stated in Equation 2, our ECP-ViT follows the conventional form of self-attention, which is softmax(QK)V.
>
>
> ### Q3  How to determine the core and peripheral components.
>
> **Response:**
>
> In our ECP-ViT, we consider the image patches as nodes and predefine a series of core ratios to divide nodes into cores and peripheries. During training, we employ Grad-CAM to identify important regions of the images and assign the core nodes to those regions. Accordingly, the QKV matrices of these patches are divided into core and peripheral components.
> For example, for images with a resolution of 224x224 and a patch size of 16x16, there are a total of 196 patch tokens as nodes. For a core ratio of 10%, around 20 patch tokens are considered as cores, and we choose the top 20 important regions as cores.
> This partitioning method is inspired by human brain networks [1], where different networks exhibit different core ratios.
> [1] "Gyri vs. sulci: Disentangling brain core-periphery functional networks via twin-transformer," MICCAI 2024.
>
>
>
>
> ### Q4 Comparison with MobileViT
>
> **Response:**
>
> Thank you for your question regarding the advantages of ECP-ViT over MobileViT. While MobileViT is a lightweight, low-latency network designed for mobile vision tasks, it achieves a top-1 accuracy of 78.4% on ImageNet. In contrast, our ECP-ViT achieves a significantly higher top-1 accuracy of 84.6% on the same dataset, demonstrating superior performance.
> Additionally, ECP-ViT employs a pruning method applied to the self-attention layer, which is an orthogonal approach to MobileViT’s design. Importantly, after pruning, each operation in ECP-ViT remains dense, making them friendly for execution on mobile GPUs. Our framework further eliminates all implicit and explicit data transformations caused by pruning and model design, which significantly reduces latency. This makes ECP-ViT highly efficient and suitable for deployment on mobile devices, offering substantial improvements in both performance and efficiency for self-attention-based transformer models.

---

> > ### Comment · Reviewer_gvXM · 2024-08-14
> >
> > Thank the authors for their effort in the rebuttal. The feedback addressed most of my concerns. I raised my initial score from 4 to 5.

---

### Author Rebuttal · Authors · 2024-08-06

### Q1 (Figure 4) How are core nodes determined?

**Response:**

In our ECP-ViT, we consider the image patches as nodes and predefine a series of core ratios to divide nodes into cores and peripheries. During training, we employ Grad-CAM to identify important regions of the images and assign the core nodes to those regions. Accordingly, the QKV matrices of these patches are divided into core and peripheral components.
For example, for images with a resolution of 224x224 and a patch size of 16x16, there are a total of 196 patch tokens as nodes. For a core ratio of 10%, around 20 patch tokens are considered as cores, and we choose the top 20 important regions as cores.
This partitioning method is inspired by human brain networks [1], where different networks exhibit different core ratios.
[1] "Gyri vs. sulci: Disentangling brain core-periphery functional networks via twin-transformer," MICCAI 2024.

### Q2 Need to explain dimension reduction heuristic algorithm.

**Response:**

Thank you for your feedback regarding the description of the dimension reduction heuristic. The detailed pseudo code for the heuristic has been attached in the uploaded PDF. The score is collected by running mini benchmarks, and the different layouts are defined by the frameworks. The process begins by identifying key nodes in the computational graph that perform actual computation (Step 1). For each key node, possible data layouts are determined (Step 2), taking into account the various layout options provided by the frameworks. Each possible layout is then evaluated by calculating a score based on data locality and GPU utilization using mini benchmarks (Step 2.1). The layout with the best score is selected and assigned to the key node (Step 3). Finally, the computational graph is updated with the new layouts for all key nodes to ensure that the entire graph benefits from the optimized layouts, improving overall performance (Step 4).

---

### Author Response · Authors · 2024-08-12
**Assistance Needed: None of the reviewers has responded**

Dear AC and SAC

We hope this message finds you well! We deeply appreciate the hard work done by the AC and SAC in the review process.

We understand that you are very busy and we aim not to add to your burden. However, all 5 of our 5 reviewers have not yet indicated that they have read our rebuttal. We would greatly appreciate your assistance in soliciting responses from the reviewers, as we are eager for feedback.

We believe our rebuttal has adequately addressed the concerns raised by all reviewers, particularly since most of the questions are simple fact-based clarification questions or requests for additional experiments to support our claims. We also want to bring to your attention that one reviewer has requested numerous additional experiments. We have devoted considerable effort to conducting these experiments at the reviewer's request, which further support our claims in the paper. However, the reviewer has not responded.

Thank you for your assistance!

Best regards,

Authors

---

### Decision · Program_Chairs · 2024-09-25

**Decision:**

Accept (poster)

**Comment:**

This paper presents ECP-ViT, a framework aimed at accelerating Vision Transformers (ViTs) for smartphone deployment. ECP-ViT enhances performance through Core-Periphery Guided Self-Attention and Data Layout Selection. The framework shows competitive results on STL-10, CIFAR-100, TinyImageNet, and ImageNet. The submission has received constructive feedback, particularly regarding the details and justification of the architecture design. We encourage the authors to carefully address the reviewers' feedback and incorporate the rebuttal into the final version.